# PathXfer: Few-Shot Visual Fidelity Transfer for Compressed Multi-to-Few Step Sampling

## Abstract

Traditional approaches to accelerate sampling in generative models rely on distillation, which requires large datasets and costly training. We instead view the quality gap between multi-step and few-step sampling as a transferable property, and introduce PathXfer, a few-shot framework that transfers multi-step fidelity to few-step sampling. PathXfer employs LoRA-based lightweight adaptation together with a Path Compression Loss, enabling effective fidelity preservation using only 16 samples, without retraining the entire model. Experiments show that PathXfer compresses sampling from 20 to 2 steps on FLUX.1-dev, a flow-based generative model, with only minor perceptual degradation, and also yields consistent improvements on diffusion models such as SDXL, demonstrating that the approach generalizes across paradigms. These results highlight few-shot fidelity transfer as a complementary approach to distillation, offering an efficient pathway for accelerating generative sampling.

## 1 Introduction

Recent advancements in generative modeling(Labs, 2024; Podell et al., 2023; Team, 2024; Han et al., 2024; Tian et al., 2024), particularly diffusion models(Rombach et al., 2022; Song et al., 2021; Ho et al., 2020; Chen et al., 2025a; Lin et al., 2024) have greatly improved the fidelity, controllability, and alignment of image synthesis. Diffusion models operate by iteratively denoising a noisy input, producing high-quality outputs but incurring substantial inference latency. Typical generation often requires 20 to 100 steps for a single image, making these models significantly slower than single-step generators like GANs (Karras et al., 2020). This high computational cost limits their deployment in latency-sensitive applications such as real-time interaction, edge devices, or high-throughput services.

To alleviate this bottleneck, prior works have explored a variety of acceleration strategies. Techniques such as custom time-step schedules (Song et al., 2021; Nichol & Dhariwal, 2021), adaptive noise reparameterization (Kingma et al., 2021; Chung et al., 2022), and caching (Zou et al., 2024; Selvaraju et al., 2024; Liu et al., 2025a; Bu et al., 2025; Liu et al., 2025b) can reduce sampling latency, but the speedup is generally modest. Methods based on knowledge distillation (Luo et al., 2023b; Salimans & Ho, 2022; Chen et al., 2025b; Zheng et al., 2024; Ren et al., 2024) can achieve larger acceleration, yet they require substantial resources: retraining student models needs large-scale datasets and extensive computation, may introduce additional architectural components, and risks degrading the fidelity of the original model. This highlights a clear trade-off between acceleration and data efficiency that existing methods struggle to resolve.

Motivated by this gap, we explore an alternative perspective: instead of treating the multi-step sampling process as a deficiency, we conceptualize the quality gap between multi-step and few-step generation as a transferable property. This property is encapsulated within the sampling path, defined as the sequence of intermediate states produced during iterative generation. Based on this insight, we introduce PathXfer, a few-shot framework that transfers high visual fidelity from the multi-step path to a compressed, few-step sampling

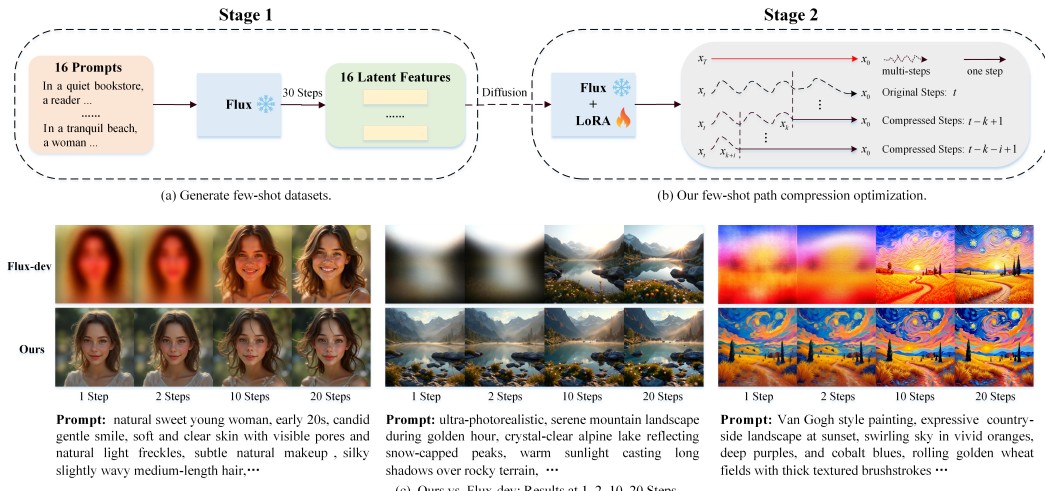

(a) Generate few-shot datasets.
(b) Our few-shot path compression optimization.

(c) Ours vs. Flux-dev: Results at 1, 2, 10, 20 Steps.

**Prompt:** natural sweet young woman, early 20s, candid gentle smile, soft and clear skin with visible pores and natural light freckles, subtle natural makeup , silky slightly wavy medium-length hair,···

**Prompt:** ultra-photorealistic, serene mountain landscape during golden hour, crystal-clear alpine lake reflecting snow-capped peaks, warm sunlight casting long shadows over rocky terrain, ···

**Prompt:** Van Gogh style painting, expressive country-side landscape at sunset, swirling sky in vivid oranges, deep purples, and cobalt blues, rolling golden wheat fields with thick textured brushstrokes ···

Figure 1: Overview of **PathXfer** for few-shot path compression. (a) Generation of 16 high-quality training examples from FLUX.1-dev, ensuring data remains within the model distribution. (b) LoRA-based adaptation and Path Compression Loss iteratively compress multi-step paths, transferring fidelity to few-step outputs. (c) Comparison across 1, 2, 10, and 20 steps showing that PathXfer effectively reduces the sampling sequence length while preserving visual quality.

process, effectively shortening the generative sequence while preserving semantic and structural fidelity (see Figure 1). PathXfer combines low-rank adaptation (LoRA) for efficient fine-tuning with a path compression loss that enforces multi-order consistency across intermediate states. During training, each target step is iteratively optimized by referencing multiple intermediate evaluations, gradually transferring the high visual fidelity of multi-step generation to a compressed, few-step path. The Path Compression Loss enables differentiable optimization without requiring teacher supervision or handcrafted schedules, and is applicable to any generative model.

In our experiments, we focus on FLUX.1-dev (Labs, 2024), a continuous-time flow-based generative model with empirically stable paths, to evaluate the core acceleration capabilities of PathXfer. Using only 16 training prompts, PathXfer compresses sampling on FLUX.1-dev from 20 to 2 steps with minimal perceptual degradation, demonstrating both efficiency and fidelity preservation (see Fig. 1c). To confirm the generality of the approach, we also evaluate PathXfer on noise-driven diffusion models such as SDXL (Podell et al., 2023) and Kolors (Team, 2024), where few-step generation quality similarly improves, illustrating that the framework can be applied across different generative paradigms.

**The main contributions of this work are summarized as follows:**

- We propose PathXfer, a few-shot lightweight fine-tuning framework that transfers visual fidelity from multi-step to few-step sampling by compressing the sequence of intermediate states for efficient acceleration.

- We introduce a Path Compression Loss based on multi-order consistency, which iteratively transfers the fidelity of multi-step generation to few-step outputs in a differentiable and model-agnostic way, enabling efficient high-quality sampling..

- We validate PathXfer on FLUX.1-dev, SDXL, and Kolors, showing that it achieves substantial acceleration with consistent fidelity preservation. These results suggest that few-shot trajectory compression provides a complementary pathway to distillation, broadening the design space for efficient generative model adaptation.

## 2 METHOD

### 2.1 BACKGROUND: SHORTEST PATH DIFFUSION

Shortest Path Diffusion (ShortDF) (Chen et al., 2025c) constructs a directed acyclic graph (DAG) $\mathcal{G}$ over discrete denoising states $x_t$ to optimize a shortcut path $\mathcal{P}^*$ that minimizes

cumulative reconstruction cost:

$$\mathcal{P}^* = \arg\min_{\mathcal{P}} \sum_{i=1}^{n} \left\| x_{0|t_i}^{\mathcal{G}} - x_{0|t_{i-1}}^{\mathcal{G}} \right\|, \quad t_{i-1} < t_i, \tag{1}$$

where $x_{0|t_i}^{\mathcal{G}}$ is the reconstructed estimate of $x_0$ from state $x_{t_i}$. ShortDF can be viewed from an ODE perspective and is trained by deterministic samplers such as DDIM (Song et al., 2021), using a dual-model self-distillation scheme in which the base model $B$ is optimized via a triangle-based relaxation of the paths from the model $\mathcal{G}$:

$$\left\| x_{0|t_i}^{B} - x_0 \right\| \approx \left\| x_{0|t_{i-1}}^{B} - x_0 \right\| + \left\| x_{0|t_i}^{\mathcal{G}} - x_{0|t_{i-1}}^{\mathcal{G}} \right\|, \tag{2}$$

Based on this relaxation, ShortDF constructs a path compression optimization strategy for multi-step sampling, accelerating inference. However, it still requires many samples and significant computational resources. Notably, intermediate multi-step paths often achieve higher visual fidelity than their corresponding shorter direct paths, motivating a lightweight ODE-based compression paradigm that transfers this fidelity along the generative trajectory to improve the shorter samping paths.

## 2.2 Visual Fidelity Transfer via Adaptive Path Compression

**A Transfer Paradigm on Generative Paths.** We introduce **PathXfer**, a novel compression method that accelerates sampling by transferring *visual fidelity* along generative trajectories. Unlike distillation, which depends on mimicking a teacher model over large datasets, PathXfer directly optimizes the ODE-based paths of a single model, making the approach both efficient and data-light.

**Path Compression via Intermediate Trajectories.** Under deterministic ODE-based sampling (such as DDIM for diffusion models (Song et al., 2021; Chen et al., 2025c) or probability flow ODE for flow matching (Labs, 2024; Liu et al., 2023)), if the reconstruction along the two-step path $t \to k \to 0$ is more accurate than the direct path $t \to 0$, the direct path can benefit from the visual fidelity of the intermediate trajectory. To implement this principle, we reformulate the triangle relation of reconstruction errors (Eq. 2) under the ODE perspective, leading to the condition:

$$\|\hat{x}_{0|t} - x_0\| > \|\hat{x}_{0|k} - x_0\|, \quad t > k, \tag{3}$$

where $\hat{x}_{0|t}$ and $\hat{x}_{0|k}$ denote reconstructions of $x_0$ from states $x_t$ and $x_k$, respectively.

To enforce this condition during training, we introduce a path compression loss for visual fidelity transfer:

$$\mathcal{L}_{\text{comp}} = \left[ \max\left(0, \|\hat{x}_{0|t} - x_0\| - \text{sg}\big(\|\hat{x}_{0|k} - x_0\|\big)\right) \right]^2, \tag{4}$$

where $\text{sg}(\cdot)$ denotes stop-gradient, preventing backpropagation through $\hat{x}_{0|k}$ and enabling the model to transfer the higher visual fidelity of the indirect path $t \to k \to 0$ to the direct path $t \to 0$. Here, the reconstruction $\hat{x}_{0|t}$ (and similarly $\hat{x}_{0|k}$) is estimated as:

$$\hat{x}_{0|t} = \begin{cases} x_t - (1 - \alpha_t) \cdot \mathbf{v}_\theta(x_t, t), & \text{(Flow Matching)} \\ \frac{1}{\sqrt{\bar{\alpha}_t}} \left( x_t - \sqrt{1 - \bar{\alpha}_t} \cdot \epsilon_\theta(x_t, t) \right), & \text{(Diffusion / deterministic sampler)} \end{cases} \tag{5}$$

where $\alpha_t$ and $\bar{\alpha}_t$ are the integration schedule coefficients, and $\mathbf{v}_\theta(x_t, t)$ and $\epsilon_\theta(x_t, t)$ are the predicted velocity and noise of the flow matching (Labs, 2024; Liu et al., 2023) and diffusion models (Ho et al., 2020; Song et al., 2021; Chen et al., 2025c), respectively, at time $t$. Note that for diffusion models, deterministic sampling is required to realize the transition from $x_t$ to $x_k$ during training, ensuring an ODE-like trajectory.

**Total Training Objective.** The total training objective combines the standard ODE-based reconstruction loss with the path compression loss:

$$\mathcal{L}_{\text{total}} = \mathcal{L}_{\text{ODE}} + \lambda \cdot \mathcal{L}_{\text{comp}}, \quad \lambda = 1 \text{ by default}, \tag{6}$$

where $\mathcal{L}_{\text{ODE}}$ is the model-specific optimization loss:

$$\mathcal{L}_{\text{ODE}} = \begin{cases} \|x_t - (x_0 + (1 - \alpha_t)\mathbf{v}_\theta(x_t, t))\|^2, & \text{(Flow Matching)} \\ \|\epsilon_\theta(x_t, t) - \epsilon\|^2, & \text{(Diffusion / deterministic sampler)} \end{cases} \quad (7)$$

This formulation allows **PathXfer** to be applied to different ODE-based generative models in a unified manner, while transferring visual fidelity along the trajectory.

**Iterative Multi-Order Compression.** The transfer process is applied iteratively during training to achieve adaptive, multi-order compression. For a given state $x_t$, we sample multiple earlier times $\mathcal{M}_t \subset \{1, ..., t-1\}$. For each $k \in \mathcal{M}_t$, the model learns to adjust its parameters such that the visual fidelity at $t$ matches or exceeds that at $k$. Ideally, after sufficient training, the learned trajectory should converge towards a state where reconstruction quality improves along the path, which can be expressed as:

$$\|\hat{x}_{0|t} - x_0\| \geq \|\hat{x}_{0|t-1} - x_0\| \geq \ldots \geq \|\hat{x}_{0|1} - x_0\|. \quad (8)$$

*Case Example.* Consider a scenario with $t = 10$ and $k = 2$. If the transfer condition holds, the direct path $x_{10} \to x_0$ inherits the visual fidelity of the two-step path $x_{10} \to x_2 \to x_0$, effectively approximating the intermediate trajectory. Next, with $t = 100$ and $k = 10$, the direct path $x_{100} \to x_0$ approximates $x_{100} \to x_{10} \to x_0$. Recursively applying this principle allows longer multi-step paths to be compressed into shorter direct paths while preserving accumulated visual fidelity, achieving second-order, third-order, or $n$-order compression dynamically.

### 2.3 Few-Shot Optimization of PathXfer with LoRA

**Few-Shot, Path-Centric Visual Fidelity Transfer with LoRA.** PathXfer accelerates sampling by transferring *visual fidelity* along a model's internal generative *paths*, specifically ODE trajectories. A key prerequisite is the *path stability* of the base model, which can vary across architectures. Here, stability means that under the same denoising path, generations from different prompts and seeds exhibit consistently similar quality. Leveraging this property, PathXfer performs fidelity transfer between different paths, making the paths themselves—rather than prompts or images—the true training samples. To adapt the model efficiently while mitigating catastrophic forgetting, we employ Low-Rank Adaptation (LoRA) (Hu et al., 2022), which inserts lightweight trainable matrices into the frozen network. This enables few-shot training to focus directly on transferring visual fidelity across generative paths, thereby preserving semantic consistency and output quality while accelerating sampling.

**Practical Implementation with FLUX.1-dev.** To accurately evaluate PathXfer, we use FLUX.1-dev (Labs, 2024), a state-of-the-art ODE-based model widely adopted in industry. FLUX is particularly suited for path compression thanks to its inherent trajectory stability, making it a representative testbed for our method. As shown in Fig. 1, we randomly select 16 prompts to generate 30-step trajectories as ground truth. Using self-generated trajectories keeps the data distribution within the model and avoids external interference. Other baselines follow the same protocol.

## 3 Experiments

**Setup**: We constructed a few-shot training dataset using 16 randomly sampled prompts from journeyDB, then generated $1024 \times 1024$ images on FLUX.1-dev with fixed noise seeds and 30 sampling steps. Training utilized $8 \times$ NVIDIA H100 GPUs (80GB) in a distributed data-parallel configuration. The DIT component of FLUX.1-dev was fine-tuned using LoRA (rank=24) with per-GPU batch size 1, totaling 18,000 iterations. Optimization employed AdamW(Loshchilov & Hutter, 2019) with hyperparameters: learning rate $2 \times 10^{-5}$, $\beta_1 = 0.9$, $\beta_2 = 0.999$, weight decay 0.01, and $\epsilon = 1 \times 10^{-8}$. Training utilized BF16 mixed precision with gradient clipping (max norm=1.0), consistent with SDXL and Kolors configurations.

**Datasets and Metrics.** Following prior work such as SANA (Xie et al., 2024), ToCache (Zou et al., 2025), and Hyper-SD (Ren et al., 2024), we evaluate on MS-COCO2017 (Lin et al., 2014) and GenEval (Ghosh et al., 2023), which contain diverse

real-world scene descriptions for comprehensive benchmarking. For MS-COCO2017, we use CLIP-Score (Radford et al., 2021) to measure image-text alignment, AES Score (Schuhmann et al., 2022) to assess visual appeal, and ImageReward (Xu et al., 2023) to evaluate overall human preference considering realism, coherence, and subjective quality. For GenEval, we use GenEval Score (Ghosh et al., 2023) to assess the model's ability to generate accurate object compositions, including relationships, positions, counts, and colors.

**Baseline Model Selection.** We select representative base models to evaluate PathXfer across architectures. For noise-based models, we choose SDXL (Podell et al., 2023) and Kolors (Team, 2024), with SDXL widely used for distillation and Kolors as an industrial open-source variant. For flow-based models, we choose FLUX.1-dev (Labs, 2024), which is highly stable, strong in quality, and naturally suitable for ODE integration, fitting our path compression and visual fidelity transfer design. FLUX.1-dev requires substantial resources, so distillation on it is less studied, with cache-based methods being more common. This choice demonstrates PathXfer's ability to deliver efficient, high-quality fidelity transfer across architectures and scenarios.

## 3.1 Main Results

**Main Results Across Models.** Table 1 compares PathXfer with state-of-the-art baselines on AES and ImageReward under varying sampling steps and training data sizes, including SDXL-Lightning (Lin et al., 2024), Hyper-SDXL (Ren et al., 2024), SDXL-LCM (Luo et al., 2023a), SDXL-TCD (Zheng et al., 2024), FLUX.1-schnell (Labs, 2024),ToCa (Zou et al., 2024), FORA (Selvaraju et al., 2024), TeaCache (Liu et al., 2025a), DiCache (Bu et al., 2025), TaylorSeer (Liu et al., 2025b).

PathXfer demonstrates broad applicability and strong acceleration across architectures. **_For noise-based models_**, using only 16 in-domain samples at 6 steps yields notable gains: SDXL's AES improves by +10.2% (5.29→5.83) and Kolors' AES improves by +5.4% (6.09→6.42) with an ImageReward gain of +31.6% (0.57→0.75). These improvements show effective few-shot visual fidelity transfer and competitiveness

Table 1: **AES and ImageReward comparison of generative models with pretrained weights**. "16" denotes 16 in-domain fine-tuning samples, "0" denotes cache methods, ">10M" denotes large-scale distillation. Bold indicates best, underlined second-best within each group.

| Model | Steps | AES ↑ | Reward ↑ | Train Data |
|---|---|---|---|---|
| **Noise-based: SDXL and Kolors at 1024×1024 resolution** | | | | |
| SDXL-Lightning | 4 | 5.63 | 0.64 | >10M |
| Hyper-SDXL | 4 | 5.74 | **0.83** | >10M |
| SDXL-LCM | 4 | 5.42 | 0.42 | 12M |
| SDXL-TCD | 4 | 5.42 | 0.59 | >10M |
| SDXL | 25 | 5.54 | 0.78 | >10M |
| SDXL | 6 | 5.29 | -0.60 | >10M |
| **Ours** | 6 | **5.83** | 0.58 | 16 |
| Kolors | 25 | 6.25 | **0.87** | >10M |
| Kolors | 6 | 6.09 | 0.57 | >10M |
| **Ours** | 6 | **6.42** | 0.75 | 16 |
| **Flow-based: FLUX.1-dev at 1024×1024 resolution** | | | | |
| ToCa | 34 | - | 1.13 | 0 |
| FORA | 34 | - | **1.20** | 0 |
| TeaCache | 15 | - | 0.97 | 0 |
| DiCache | 13 | - | -0.65 | 0 |
| TaylorSeer | 10 | - | 1.00 | 0 |
| FLUX.1-dev | 20 | 5.82 | 1.19 | >10M |
| FLUX.1-dev | 10 | 5.57 | 0.93 | >10M |
| FLUX.1-dev | 4 | 5.17 | -0.25 | >10M |
| FLUX.1-schnell | 4 | 5.57 | 1.13 | large-scale |
| **Ours** | 4 | **6.00** | 1.11 | 16 |
| **Ours** | 2 | 5.99 | 1.07 | 16 |
| **Ours** | 1 | 5.94 | 1.03 | 16 |

with distillation-based baselines while offering distillation-like acceleration advantages. **_For flow-based FLUX.1-dev_**, PathXfer achieves even greater improvements. At 4 steps, AES increases by +16.1% (5.17→6.00). PathXfer maintains strong performance (AES 5.99 at 2 steps, 5.94 at 1 step) while enabling up to a 10×–20× speedup compared to the 20-step baseline. PathXfer achieves comparable quality with strong competitiveness to distillation-based methods and, compared to cache-based approaches, offers significant speed advantages while retaining a light-weight design that only requires minimal in-domain samples. These results confirm PathXfer's general feasibility. The stronger gains on FLUX.1-dev stem from its intrinsic ODE properties, which align well with our path compression and visual fidelity transfer strategy, whereas noise-based models require deterministic samplers such as DDIM to approximate ODE-like behavior. This motivates focusing subsequent experiments on FLUX.1-dev for fast and effective validation.

**GenEval comparison in FLUX.1-dev.** Table 2 reports GenEval scores as a comprehensive quantitative evaluation across different generative models. The top group lists representative cross-architecture SOTA baselines, including SDXL, SD3-Medium (Esser et al., 2024), SANA (Xie et al., 2024), PCM (Wang et al., 2024), SD3.5-Turbo (Esser et al., 2024), SDXL-Turbo (Podell et al., 2023), and SDXL-DMD2 (Yin et al., 2024), serving as global references. The bottom group presents a direct comparison under the FLUX.1 architecture.

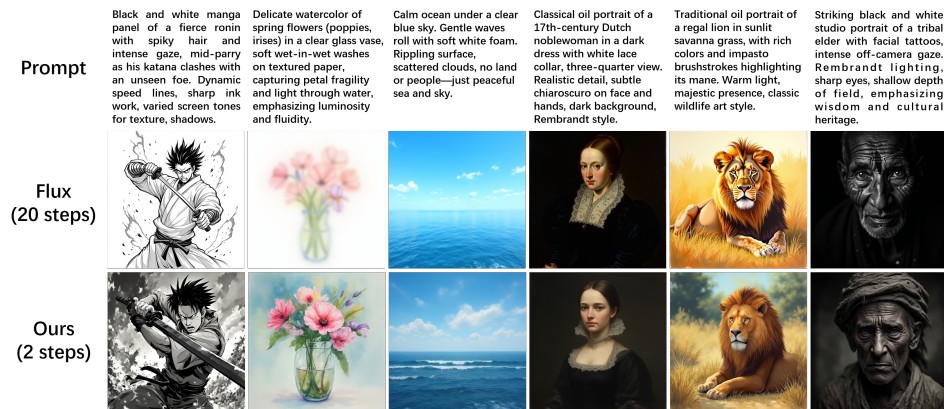

Figure 2: Comparison with FLUX.1-dev (20 steps) shows our 2-step method achieves sharper, more realistic results across diverse scenes. (see Appendix Fig. 5 for more results).

PathXfer, trained with only 16 in-domain samples, consistently improves both efficiency and quality over FLUX.1-dev. At 1 step, PathXfer reaches 0.604 GenEval, closely approaching the 20-step FLUX.1-dev baseline (0.648) while reducing trajectory length nearly tenfold, demonstrating substantial speedup. At 10 and 4 steps, PathXfer also achieves significant gains (0.649 and 0.612, respectively), further demonstrating its effectiveness. Compared to FLUX.1-schnell (Labs, 2024) at 4 steps, which uses large-scale distillation, PathXfer delivers comparable quality with far fewer samples, showing that it can rival distillation-based approaches under extreme compression.

**Qualitative Results.** Fig. 2 and Fig. 3 present complementary visual comparisons for PathXfer. Fig. 2 shows PathXfer (2 steps) versus FLUX.1-dev (20 steps), where despite using only a tenth of the steps, PathXfer consistently produces sharper, more coherent images across diverse categories such as humans, animals, architecture, and abstract art. It better preserves structural boundaries, texture details, and global composition, while reducing blurring and semantic drift under extreme compression. Fig. 3 provides a targeted comparison between PathXfer (4 steps) and FLUX.1-schnell (4 steps distilled), highlighting superior semantic preservation and structural

Table 2: **GenEval comparison on FLUX.1-dev. Bold** indicates the best and underlined the second-best within each group.

| Method | Steps | GenEval ↑ | Train Data |
|---|---|---|---|
| **SOTA Methods** | | | |
| SDXL | 50 | 0.550 | >10M |
| SD3-Medium | 28 | 0.620 | >10M |
| SANA | 20 | 0.66 | >10M |
| PCM† | 4 | 0.560 | 12M |
| SD3.5-Turbo | 2 | 0.530 | >10M |
| SDXL-Turbo | 1 | 0.510 | >10M |
| SDXL-DMD2 | 1 | 0.590 | 500K |
| **Direct Comparison (FLUX.1 Architecture)** | | | |
| FLUX.1-schnell | 4 | **0.710** | large-scale |
| FLUX.1-dev | 20 | 0.648 | |
| FLUX.1-dev | 10 | 0.545 | >10M |
| FLUX.1-dev | 4 | 0.296 | |
| **Ours** | 10 | 0.649 | |
| **Ours** | 4 | 0.612 | **16** |
| **Ours** | 1 | 0.604 | |

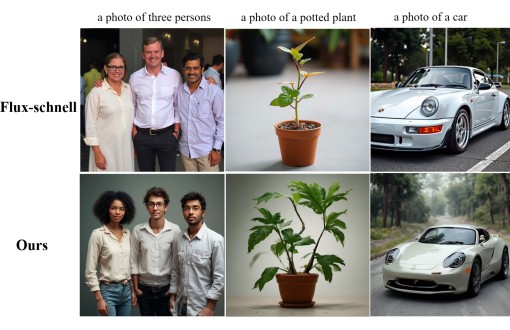

Figure 3: **4-step comparison.** Ours avoids artifacts and yields better structure, texture, and shadow consistency than FLUX.1-schnell.

integrity in PathXfer's outputs. It avoids common artifacts such as hallucinated limbs, unnatural object placements, or distorted proportions, and maintains high visual coherence across diverse categories (e.g., humans, vehicles, fine textures). These observations confirm PathXfer's capability as a practical few-shot fidelity transfer method, achieving competitive results with minimal data and computation even against heavily distilled baselines.

Together with the **User Study** results in Appendix B, which further validate perceptual quality from a human perspective, these findings confirm that PathXfer offers an effective

Table 3: **Unified ablation study on LoRA, rank, and few-shot data size.** We compare the effect of LoRA, LoRA rank, and training prompt count across different sampling steps. Our method is robust to extreme compression and generalizes well with as few as 16 training samples.

| Step | LoRA Setting | #Prompts | CLIP ↑ | AES ↑ | Reward ↑ | GenEval ↑ |
|------|-------------|----------|--------|-------|----------|-----------|
| 1 | w/o LoRA | 16 | 22.24 | 4.16 | -2.14 | 0.05 |
| | rank 8 | 16 | 28.37 | 5.06 | -0.94 | 0.20 |
| | rank 24 | 16 | 31.95 | 5.94 | 1.03 | 0.60 |
| | rank 24 | 24 | 32.10 | 5.90 | 1.04 | 0.57 |
| | rank 24 | 32 | 31.62 | 5.94 | 0.94 | 0.55 |
| 2 | w/o LoRA | 16 | 24.07 | 4.58 | -1.97 | 0.10 |
| | rank 8 | 16 | 30.66 | 5.28 | -0.03 | 0.33 |
| | rank 24 | 16 | 32.00 | 5.99 | 1.07 | 0.60 |
| | rank 24 | 24 | 31.99 | 5.98 | 1.08 | 0.58 |
| | rank 24 | 32 | 31.87 | 5.95 | 1.04 | 0.60 |
| 4 | w/o LoRA | 16 | 30.81 | 5.26 | 0.30 | 0.37 |
| | rank 8 | 16 | 30.29 | 5.68 | 0.84 | 0.49 |
| | rank 24 | 16 | 31.77 | 6.00 | 1.11 | 0.61 |
| | rank 24 | 24 | 32.04 | 5.95 | 1.15 | 0.61 |
| | rank 24 | 32 | 32.05 | 5.98 | 1.13 | 0.62 |
| 10 | w/o LoRA | 16 | 31.57 | 5.53 | 0.94 | 0.54 |
| | rank 8 | 16 | 31.15 | 5.80 | 1.28 | 0.67 |
| | rank 24 | 16 | 31.95 | 6.05 | 1.22 | 0.65 |
| | rank 24 | 24 | 32.11 | 5.99 | 1.24 | 0.65 |
| | rank 24 | 32 | 31.93 | 6.00 | 1.20 | 0.63 |

Table 4: **Comparison of performance and inference efficiency under trajectory compression.** Our method maintains or improves CLIP and AES scores while significantly accelerating inference. Gains are reported as relative improvements over FLUX.1-dev.

| Steps | CLIP Score ↑ | CLIP Gain (%) | AES Score ↑ | AES Gain (%) | Latency (s) ↑ | Speed-Up |
|-------|--------------|---------------|-------------|--------------|---------------|----------|
| | FLUX.1-dev / Ours | | FLUX.1-dev / Ours | | FLUX.1-dev / Ours | |
| 50 | 32.33 / 31.89 | -1.36% | 5.85 / 6.00 | +2.56% | 19.94 / 10.57 | 1.89× |
| 30 | 32.36 / 31.89 | -1.45% | 5.85 / 6.02 | +2.91% | 11.62 / 6.42 | 1.81× |
| 20 | 32.31 / 31.91 | -1.24% | 5.82 / 6.03 | +3.61% | 7.45 / 4.34 | 1.72× |
| 10 | 31.94 / 31.95 | +0.03% | 5.57 / 6.05 | +8.62% | 3.30 / 2.27 | 1.45× |
| 4 | 28.11 / 31.77 | +13.0% | 5.18 / 6.00 | +15.83% | 1.03 / 1.03 | 1.00× |
| 2 | 24.07/ 32.00 | +32.95% | 4.53 / 5.99 | +32.23% | 0.61 / 0.61 | 1.00× |
| 1 | 22.25 / 31.95 | +43.59% | 4.11 / 5.94 | +44.53% | 0.41 / 0.41 | 1.00× |

trajectory compression strategy, enabling reliable fidelity transfer with minimal data and computation, even in highly constrained sampling settings.

### 3.2 ABLATIONS

**Unified Analysis of LoRA, Rank, and Few-shot Generalization.** Table 3 presents a unified ablation study on the effects of LoRA, LoRA rank, and training prompt count across different steps. The results yield three key observations: (1) **LoRA is critical under compression.** Without LoRA, performance drops sharply, especially at 1–2 steps where GenEval falls below 0.1 and ImageReward turns negative. Adding LoRA significantly boosts all metrics (e.g., GenEval: 0.05 → 0.60, Reward: –2.14 → 1.03 at 1 step). (2) **Higher rank improves quality.** Increasing LoRA rank from 8 to 24 consistently enhances performance, particularly for CLIP and Reward (e.g., at 2 steps, CLIP: 30.66 → 32.00, Reward: –0.03 → 1.07), suggesting that higher-rank adapters better capture velocity field refinements. (3) **Few-shot generalization is strong.** With only 16 prompts, our model achieves solid performance across all steps. Using more prompts (24/32) yields little or no gain, especially in ACE and GenEval, indicating effective generalization from minimal data. These findings confirm that LoRA enables robust performance under compression, and strong few-shot generalization ensures efficiency and practicality.

**Compression vs. Quality Trade-off.** Table 4 summarizes the performance and efficiency of our method under different trajectory compression levels. Despite drastically reducing the number of sampling steps, our method consistently preserves or even improves generation

quality while significantly accelerating inference. In high-step regimes (50–30 steps), our method achieves 1.8×–1.9× speed-ups over FLUX.1-dev, with AES scores steadily improving (e.g., +2.56% at 50 steps), and only minimal CLIP drop (<1.5%). At 10 steps, we match CLIP and significantly improve AES (+8.62%), while reducing latency by 31%. Notably, in low-step settings (≤ 4 steps), our method not only achieves real-time speed (0.4–1.0s), but also shows large gains over FLUX.1-dev in both CLIP and AES—up to +44.5% AES and +43.6% CLIP at 1 step. This reversal is due to FLUX.1-dev collapsing under compression, while our model remains stable and effective. These results demonstrate that our trajectory compression framework not only accelerates generation but also enhances perceptual and semantic quality, especially under extreme inference constraints.

**Quality Retention under Step Compression.** Table 5 demonstrates that our method consistently retains or improves generation quality (measured by AES) while significantly accelerating inference. Using FLUX.1-dev at 30 steps as the baseline (AES: 5.84, 1.0× speed), our method not only surpasses this quality baseline at every compressed step count, but also offers large speed-ups. Notably, even when reducing to just 10 steps, our method improves AES to 6.05 (+3.60%) while achieving a 5.12× speed-up. Further compression to 2 steps still yields 5.99 AES (+2.57%) with a remarkable 19.05× acceleration. This quality-preserving trend

Table 5: **Quality retention with step compression.** Our method sustains competitive quality while delivering substantial speed-ups. *Baseline* refers to FLUX.1-dev with 30 steps (AES = 5.84), and all gains are reported relative to this baseline.

| Steps | AES score ↑ | Speed-up |
|---|---|---|
| *Baseline* | 5.84 | 1.00× |
| 30 | 6.02 (+3.08%) | 1.81× |
| 20 | 6.03 (+3.25%) | 2.68× |
| 10 | 6.05 (+3.60%) | 5.12× |
| 4 | 6.00 (+2.74%) | 11.28× |
| 2 | 5.99 (+2.57%) | 19.05× |

holds steadily across all compression levels, confirming the stability of our trajectory modeling under aggressive step reduction.

**Validation of Path Stability through Velocity Analysis.** Figure 4 shows estimated velocity $v$ over a 50-step sampling trajectory. The original *FLUX.1-dev* exhibits early-step fluctuations that gradually smooth out, indicating unstable trajectories. In contrast, PathXfer maintains consistently low velocity variation, demonstrating smoother and more predictable generative paths. This empirically supports the stage-wise monotonicity constraint in Eq. 8 and validates our multi-order compression loss. Such stability underlies PathXfer's strengths: it enables effective fidelity transfer

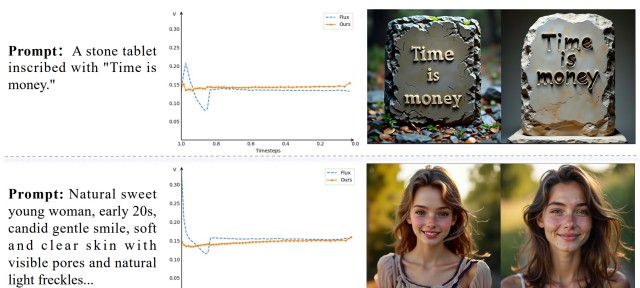

Figure 4: Estimated velocity $v$ over 50 steps to assess trajectory stability. FLUX.1-dev exhibits early-stage fluctuations, whereas our method maintains consistently low variation across steps, indicating a smoother sampling path that supports the path compression strategy and aligns with the stage-wise continuity constraint in Eq. 8.

across models and domains, allowing few-shot adaptation while preserving style-level and structural diversity (Appendix. C). This confirms that the "training samples" in PathXfer capture trajectory priors rather than being tied to specific prompts or images, and that the core of few-shot generalization lies in stable path modeling. Together with the quantitative gains and improvements, this validates that path stability is fundamental for achieving efficient, high-fidelity, and diverse generation under extreme compression.

**Extension to LoRA Fusion.** To further illustrate the plug-and-play capability of PathXfer, we integrate our compressed model with style-specific LoRAs from the community (e.g., *Neza* (Stefan, 2025), *Sketch* (Xia, 2025)) which are open-sourced FLUX.1-dev LoRAs downloaded from the community. As detailed in Appendix E, PathXfer enables 4-step sampling to achieve style-preserving results comparable to 20-step generation, demonstrating seamless compatibility with diverse LoRA extensions.

## 4 Related Work

**Text-to-Image Diffusion Models.** Text-to-image diffusion models (Rombach et al., 2022; Podell et al., 2023; Esser et al., 2024; Labs, 2024) have become the dominant framework for high-quality image synthesis. Stable Diffusion (SD) (Rombach et al., 2022) first enabled practical large-scale generation by operating in a latent space. SDXL (Podell et al., 2023) improves fidelity through a deeper UNet and dual text encoders, while SD3 (Esser et al., 2024) introduces rectified flow and cross-modal transformers for better compositionality. In contrast to these discrete-step methods, Flux (Labs, 2024) models generation as a continuous-time probability flow ODE, learning a velocity field via flow matching and enabling smooth, step-free sampling. Despite strong performance, these models typically require 20-100 iterative steps for high-quality output, leading to substantial inference latency that hinders real-time applications. This limitation motivates the need for principled, trajectory-level acceleration strategies that preserve fidelity while reducing sampling depth.

**Sampling Optimization Strategies.** Various strategies have been proposed to reduce the inference cost of generative models (Song et al., 2024; Ren et al., 2024; Chen et al., 2025a; Zou et al., 2025), including knowledge distillation, timestep reduction, and feature reuse. Latent Consistency Models (LCMs) (Luo et al., 2023a) reformulate reverse diffusion as latent-space probability flow ODEs to achieve 2-4 step high-fidelity generation. DeepCache (Ma et al., 2024) leverages temporally stable U-Net features for caching, while ParaTAA (Tang et al., 2024) uses fixed-point iteration and parallel sampling for up to $400\times$ acceleration. Hyper-SD (Ren et al., 2024) unifies multiple consistency-based distillations to support 1-8 step generation, and Skip-DiT (Chen et al., 2025a) and ToCa (Zou et al., 2025) introduce long-skip connections and token-wise adaptive caching. Although these methods achieve notable acceleration, those that require minimal data or computation typically provide limited speedup, while methods that achieve large acceleration often rely on extensive training datasets or heavy computation. Achieving efficient high-fidelity generation with low resource requirements remains challenging.

**Few-Shot and LoRA Fine-Tuning Mechanisms.** Recent works explore few-shot LoRA for style transfer, achieving visual alignment with as few as 3-10 images. For example, StyleLoRA (Gal et al., 2023) learns identity-preserving edits from a handful of reference images, while DragStyle (Shi et al., 2023) enables spatially controllable stylization under few-shot constraints. Complementarily, parameter-efficient tuning methods aim to adapt large diffusion models to new tasks with minimal updates. LCM-LoRA (Luo et al., 2023b) provides plug-and-play adaptation via lightweight modules distilled from Latent Consistency Models. Hyper-SD (Ren et al., 2024) proposes a unified LoRA design supporting multi-step generation under reduced sampling budgets, further enhanced by human preference learning. SANA-Sprint (Chen et al., 2025b) introduces a step-adaptive architecture trained across 1-4 steps and integrated with ControlNet for real-time generation. However, the intersection of few-shot tuning and sampling acceleration remains underexplored, offering an opportunity to leverage trajectory stability for efficient path compression with minimal effort.

## 5 Conclusion

We present **PathXfer**, a few-shot framework for transferring visual fidelity from multi-step to few-step sampling. By combining a path compression loss with lightweight LoRA adaptation, PathXfer compresses generative paths while preserving perceptual and semantic quality. Experiments on FLUX.1-dev, as well as validation on SDXL and Kolors, demonstrate that it generalizes across both flow- and diffusion-based models, suggesting that few-shot fidelity transfer can serve as a promising, data-efficient complement to distillation for accelerating generative sampling. While PathXfer provides a broadly applicable approach to fast, high-quality generation, challenges remain for highly diverse or multi-modal prompts, where maintaining path smoothness is more difficult, and the current design assumes access to high-quality final samples for supervision. Future work will explore dynamic step allocation, adaptive compression based on prompt complexity, and unsupervised path refinement to further improve robustness and generalization.

## 6 ETHICS STATEMENT

According to the author guidelines, we include this content, which does not count toward the total page limit. We are committed to conducting research on PathXfer in a responsible and ethical manner, ensuring adherence to high standards of integrity. Our work follows institutional ethical guidelines and complies with relevant protocols for data collection, processing, and dissemination. In applying generative models, we take care to ensure that generated content does not infringe upon the rights of individuals or communities. Measures are taken to reduce the risk of misuse, such as the creation of misleading or harmful content, and we emphasize the importance of responsible use. We also acknowledge the broader societal implications of advanced generative technologies. While PathXfer aims to enhance creative and efficient generation, we remain attentive to ethical concerns including privacy, consent, and potential misuse. We advocate for deployment with appropriate safeguards to protect rights and dignity. All datasets used in our research comply with privacy regulations and ethical standards, and are sourced appropriately.

## 7 REPRODUCIBILITY STATEMENT

According to the author guidelines, we include this content, which does not count toward the total page limit. To support reproducibility and broader research impact, we plan to make key resources related to PathXfer available to the community, including code, datasets, and model weights where permissible. Implementation details and relevant assets will be shared in a manner that balances openness with applicable usage considerations. This approach aims to facilitate verification of our results and encourage further exploration of PathXfer while respecting practical constraints

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

# PathXfer: Few-Shot Visual Fidelity Transfer for Compressed Multi-to-Few Step Sampling (Appendix)

This appendix provides detailed supplementary materials to support our main paper, including pseudocode for PathXfer, user study results, diversity analysis, noise-based results such as Kolors results, style-preserving LoRA fusion experiments, comparison between PathXfer and knowledge distillation, and intermediate test samples. Together, these materials validate the efficiency, fidelity preservation, diversity, and scalability of our proposed method. We also describe the role of Large Language Models (LLMs) in this work in Appendix H.

## A PSEUDOCODE

To enhance comprehension of the path compression optimization process, we present the following PyTorch-like pseudocode (1), which elegantly outlines the essential steps of our training procedure. This includes the stepwise reconstruction error estimation for timesteps $t$ and $k$, as well as the loss optimization pivotal to achieving path compression. We will release the source code after the review process is completed.

```python
for epoch in range(num_train_epochs):
    for step, batch in enumerate(train_dataloader):
        x_0 = batch['x_0'] # Sample, latent space
        noise, prompts = batch['noise'], batch['prompts']

        # Randomly sample two timesteps t and k, k < t
        t = torch.randint(1, 1000, size=(x_0.size(0),))
        k = (torch.rand(size=t.shape) * t).to(t)
        t, k = get_schedule(t), get_schedule(k)

        # Add noise
        x_t = (1 - t) * x_0 + t * noise
        # predict the reconstruction error at timestep t
        v_t = model(x_t, t, prompts) # velocity v at timestep t
        x0_t = x_t - t * v_t # the prediction of x0 at timestep t
        dt = (x_0 - x0_t).abs() # reconstruction error

        # Similarly, the reconstruction error at timestep k
        with torch.no_grad():
            x_k = x_t - (t - k) * v_t  # from t to k
            v_k = model(x_k, k, prompts)
            x0_k = x_k - k * v_k
            dk = (x_0 - x0_k).abs()

        # Optimization
        flow_loss = F.mse_loss(v_t, noise - x_0)
        relax = dt > dk #Relaxation condition
        relax_loss = relax * (dt - dk) ** 2
        loss = flow_loss.mean() + relax_loss.mean()
        optimizer.zero_grad()
        loss.backward()
        optimizer.step()
```

Listing 1: PyTorch-like training loop with our path compression optimization

## B USER STUDY

We conducted a user study to compare the performance of our method with the original FLUX.1-dev model. A total of 24 participants were involved, including professionals in computer science, individuals with expertise in aesthetics, and general users. The study focused on two main aspects of image generation: **Semantic Matching** (how well the image

Figure 5: Examples comparing images generated by our method (2-step, left) and the original FLUX.1-dev model (20-step, right). Across a range of prompts, our method produces results that are visually and semantically close to those of FLUX.1-dev, despite using far fewer inference steps.

reflects the given prompt) and **Overall Image Quality** (including clarity, aesthetic appeal, and personal preference). For each aspect, 816 pairwise comparisons were collected, resulting in 1632 judgments. Participants were shown image pairs—one generated by our 2-step method and the other by the 20-step FLUX.1-dev model—with their positions randomized to reduce bias.

Table 6: User study results comparing our method (2-step) with FLUX.1-dev (20-step). Each score indicates the number of votes received out of 816 total comparisons per category.

| Method | Semantic Matching | Overall Image Quality |
|---|---|---|
| Ours (2-step) | 454 | 474 |
| FLUX.1-dev (20-step) | 362 | 342 |

Table 6 summarizes the results. Our method received more votes in both categories: 454 vs. 362 for semantic matching, and 474 vs. 342 for image quality. Despite using only a fraction of the inference steps, our approach was consistently preferred. As shown in Figure 5, the outputs of our method are often similar in both content and visual quality to those generated by the 20-step FLUX.1-dev model. While the number of qualitative examples is limited, user feedback suggests that in some cases, the 2-step results were even preferred. These findings indicate that our method provides a strong trade-off between quality and efficiency, making it suitable for scenarios where speed and visual fidelity are both important.

## C  Diversity Analysis

To evaluate style-level and structural diversity under extreme compression, we conduct experiments using LPIPS (VGG backbone) over selected MS-COCO prompts, each with 10 random seeds. Style diversity is ensured via prompt engineering, while structural/content diversity is quantified by averaging LPIPS scores. Table 7 summarizes the results.

Table 7: Comparison of diversity results between our method (2-step) and FLUX.1-dev (20-step).

| Method | LPIPS (VGG) |
|---|---|
| Ours (2-step) | 0.728 |
| FLUX.1-dev (20-step) | 0.592 |

Table 7 shows that our PathXfer (2-step) produces higher LPIPS scores compared to FLUX.1-dev (20-step), indicating greater diversity preservation even under strong trajectory compression. This suggests that our method is not only efficient but also capable of maintaining a rich variety of outputs.

Figures 2 and 3 further support these findings. Fig. 2 provides a broad qualitative comparison between PathXfer and FLUX.1-dev, illustrating that PathXfer preserves sharper structures and better semantic consistency under a 2-step setting compared to the 20-step baseline. Fig. 3 presents a targeted comparison between PathXfer's 4-step outputs and FLUX.1-schnell's 4-step distilled outputs, highlighting that PathXfer better preserves semantic correctness and structural integrity while avoiding common artifacts such as hallucinated limbs or unnatural object placements.

As an additional case study, we examine a complex prompt (used for Fig. 1):

```
"natural sweet young woman, early 20s, candid gentle
smile, soft clear skin with visible pores and natural
light freckles, subtle natural makeup, silky slightly wavy
medium-length hair, fresh lively expression, background shows
a park with trees and a small lake/stream, captured by a
Leica SL2-S with Summilux-M 50 mm f/1.4 lens, natural outdoor
sunlight (golden hour), cinematic warm tones, slight wind
blowing hair, extremely detailed facial texture, true-to-life
color grading, authentic human proportions, real-world
photography style."
```

This prompt is a detailed description of a young woman scene containing multi-entity composition, fine-grained attributes, spatial reasoning, and cross-domain cues. PathXfer successfully preserves scene coherence, fine details, and photographic realism even under a compressed 2-step setting. Its ability to maintain visual coherence across diverse categories (e.g., humans, vehicles, fine textures) indicates that it learns transferable priors rather than overfitting to prompt-specific samples, thus validating its few-shot capability.

Overall, these quantitative and qualitative analyses suggest that PathXfer offers a promising balance between fidelity, diversity, and efficiency. The diversity evaluation in Table 7 and the qualitative comparisons in Figures 2 and 3 jointly demonstrate that our path compression strategy maintains rich, coherent generation even under extreme step reduction, supporting its potential for practical low-resource deployment.

## D  Kolors Results

As illustrated in Figure 6, we compare the 10-step outputs of our method with the 20-step outputs of the Kolors model, a representative commercial open-source model under the SDXL architecture. Including Kolors' results serves to further demonstrate the effectiveness of PathXfer across diffusion model families. While both approaches generate images of comparable visual quality, our method demonstrates two clear advantages. First, it achieves

similar fidelity with only half the inference steps, highlighting its efficiency. Second, and more importantly, our results exhibit stronger semantic alignment with the input prompts: the highlighted elements (marked in red) are faithfully reflected in our generations, whereas Kolors often fails to capture them.

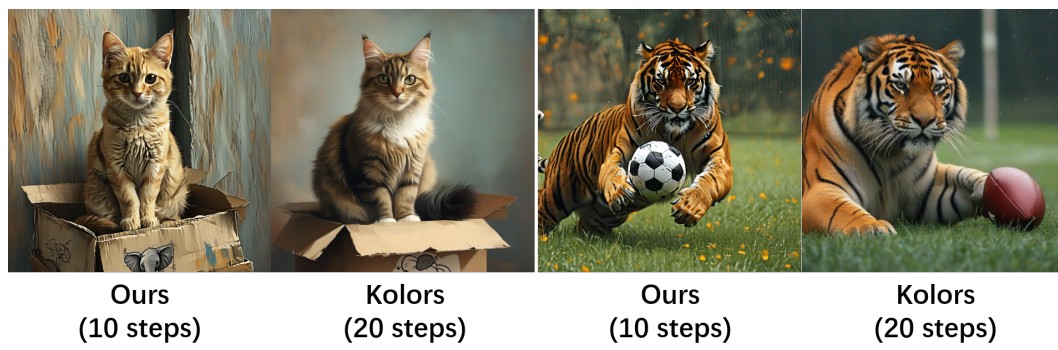

Figure 6: Examples comparing images generated by our method (10-step, left) and the original Kolors model (20-step, right). While both methods produce visually comparable results, our approach more faithfully captures the highlighted semantic elements in the prompts (marked in red), which are often missed or only partially reflected by Kolors.

## E   Style-Preserving Path Compression via LoRA Fusion

Figure 7 illustrates the impact of our PathXfer LoRA fusion compared to baseline LoRA settings. The figure compares the original FLUX.1-dev with LoRA at 4-step and 20-step generation, as well as results after fusing with open-source style LoRAs such as Neza and Sketch. In the original LoRA models, the 4-step generation is not viable for practical use due to noticeable quality, while the 20-step generation yields acceptable results. After applying our PathXfer fusion, the 4-step generation successfully retains the style fidelity and quality of the 20-step baseline.

This demonstrates that our path compression LoRA can serve as a plug-and-play tool, seamlessly integrating into various style LoRAs of the base model without compromising quality. The approach not only preserves original style characteristics but also significantly enhances scalability and applicability, enabling efficient low-step generation across diverse style variants.

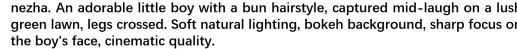

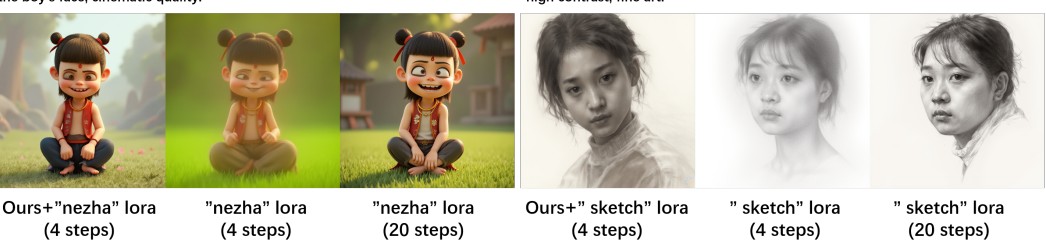

Figure 7: Style-preserving results via PathXfer LoRA fusion. Compared to FLUX.1-dev with 4-step and 20-step generation, fusing PathXfer with style LoRAs such as Neza and Sketch enables the 4-step generation to match the quality and style fidelity of the original 20-step outputs. This demonstrates the plug-and-play capability and scalability of our path compression LoRA.

# F  COMPARISON BETWEEN PATHXFER AND KNOWLEDGE DISTILLATION

PathXfer differs fundamentally from conventional Knowledge Distillation (KD). KD relies on an external teacher model to supervise a student model. In contrast, PathXfer requires no additional teacher. Instead, it directly references the results of indirect generative paths of the target model itself. Through iterative updates on few-shot examples, PathXfer effectively approximates the optimization of long paths. This can be viewed from a recursive perspective, though it is not strictly recursion, but rather a **path-stability-based boot-strapping** strategy. The core idea is to use the target model's own high-quality generation paths as training signals, enabling accelerated sampling while preserving visual fidelity and diversity.

Table 8 summarizes the differences between PathXfer and KD, illustrating the distinct supervision sources, data requirements, and optimization goals.

Table 8: Comparison of PathXfer and Knowledge Distillation

| Feature | PathXfer | Knowledge Distillation (KD) |
|---|---|---|
| Supervision Source | Self-generated paths (direct referencing of indirect paths) | External teacher model |
| Bootstrapping | Yes | No |
| Data Requirement | Few-shot examples | Large-scale dataset |
| Optimization Goal | Path compression + fidelity preservation | Student model approximation of teacher |
| Application Scenarios | Multi-step compression, low-resource settings | High-quality model training |

In summary, as shown in Table 8, PathXfer offers a self-bootstrapping, path-centric approach that leverages the target model's own generative trajectories. This enables efficient multi-step compression with minimal additional supervision, offering a complementary approach to knowledge distillation that is lightweight, adaptable, and suited for low-resource scenarios.

# G  INTERMEDIATE TEST SAMPLES

During the training process, we perform random testing every 1,000 iterations to facilitate real-time monitoring of the path compression effect. In these tests, the number of denoising steps is randomly selected from 1 to 10, and the generated images are of 512-pixel size to expedite the process. The results of these tests are available in the "**intermediate test samples**" folder of the supplementary materials. From these test results, it can be observed that our results become increasingly clear and accurate compared to the original FLUX.1-dev.

# H  LLM USAGE

Large Language Models (LLMs), specifically GPT-5, were used solely to assist with grammar correction, sentence refinement, and improving the clarity and coherence of the Introduction and Method sections. All scientific content, experimental design, and conclusions were determined, verified, and critically edited by the authors. The LLM contribution did not involve generating factual content, experimental results, or mathematical derivations.

