# OpenReview forum: "PathXfer: Few-Shot Visual Fidelity Transfer for Compressed Multi-to-Few Step Sampling"
_ICLR.cc/2026/Conference — ICLR 2026 Conference Withdrawn Submission_

### Official Review · Reviewer_vEqK · 2025-10-26

**Soundness:** 2
**Presentation:** 3
**Contribution:** 3
**Rating:** 6
**Confidence:** 3

**Summary:**

The paper proposes PathXfer, which proposes to accelerate the sampling of diffusion and flow models by a path compression loss. The loss enforces the model's few step sampling to match or exceed its multi step sampling path. Crucially, the approach is highly efficient, requiring only 16 training samples and only on LORA weights. Empirical results show the method to be effective at few step sampling while preserving image quality.

**Strengths:**

- Highly efficient and lightweight method compared to distillation methods, using only 16 samples and LORA and teacher-free.
- The proposed training objective is conceptually clean -- enforce that earlier steps match or exceed later steps at reconstruction.
- Paper is overall well-written and clear.

**Weaknesses:**

- Some theoretical concerns that could be addressed: first, the paper assumes that for $k<t$, the construction from $k$ will be uniformly better and thus serves as a training signal. Is this a true assumption? Empirically it seems to hold, but generative samplers can be non-monotic.

- Second, by using the model as its own 'teacher', it seems to be a moving target self-distillation objective, which could be inherently unstable. Can the authors show that the loss will converge under this dynamic?

- It will be helpful to have exact compute/time comparisons to support training efficiency claims on smaller models/datasets like ImageNet models.

- In Table 1, PathXfer models produce strong AES results but consistently lower Reward score. Can the authors discuss the reasons for this and the implications?

- Qualitative results is subjective; for instance, I find the results of Flux-schnell in Fig 3 generally more appealing though there are some hallucinations in the persons image. There will also be some degree of cherry picking in presenting qualitative results. Including qualitative results is fine, but I advise the authors to tone down the claims made off them in the paragraph lines 302-320.

**Questions:**

- Which ODE samplers is used for the diffusion model experiments, and have the authors explored whether higher-order ODE samplers lead to improved performance over something like a first-order DDIM?
- Have the authors studied the robustness of the method to different 16 prompts used for generating the training images?
- For the same number of steps, why does PathXfer offer speedups over the base model in Table 4?

---

### Official Review · Reviewer_Q3cZ · 2025-10-30

**Soundness:** 2
**Presentation:** 3
**Contribution:** 2
**Rating:** 2
**Confidence:** 4

**Summary:**

This paper presents PathXfer, which transfers multi-step fidelity to few-step sampling via LoRA-based adaptation and Path Compression Loss, achieving compression with only 16 samples and minimal quality loss across models.

**Strengths:**

1. Provides a lightweight, model-agnostic few-shot adaptation framework that uses LoRA fine-tuning on frozen diffusion backbones. The path-compression loss is simple, differentiable, and architecture-independent, allowing integration across pretrained models such as FLUX, SDXL, and Kolors with only 16 in-domain samples.

2. Achieves up to 10×~20× faster sampling speed while maintaining comparable perceptual quality and prompt fidelity, demonstrating clear efficiency gains without altering core model architectures.

3. Maintains high perceptual quality and computational efficiency using lightweight LoRA, preserving FID, CLIP, and ImageReward scores across large-scale diffusion pipelines while introducing negligible parameter overhead.

**Weaknesses:**

1. The theoretical and mathematical foundations of the ODE reconstruction equations and the path consistency objective are nearly identical to those of Shortest Path Diffusion(ShortDF)[1]. PathXfer primarily reuses these formulations under a few-shot LoRA fine-tuning context rather than introducing a fundamentally new mechanism. Given this extensive reuse of ShortDF’s structure and objective, PathXfer’s actual contribution appears limited to engineering simplification through LoRA-based few-shot fine-tuning and its application to larger diffusion pipelines.

[1] Optimizing for the shortest path in denoising diffusion model. (CVPR 2025)

2. The “Path Compression Loss” does not model the composite path $t \rightarrow k \rightarrow 0$, but merely enforces monotonic improvement between $k \rightarrow 0$ and $t \rightarrow 0$. In Eq(3), $\hat{x}_{0|t}$ and $\hat{x}_{0|k}$ denote reconstructions of ${x}_{0}$ from states ${x}_{t}$, ${x}_{k}$, respectively. In ShortDF, the transition $t \rightarrow k$ is explicitly realized through the edge weight formulation, where the estimate $\hat{x}'_{0|k}$ is computed using $\hat{x}_k$, while PathXfer contains no composite path $t \rightarrow k$ in its computational graph at all. This weakens its conceptual grounding as a true path consistency method and reduces interpretability of “multi-order compression”.

3. Reported metrics (AES, ImageReward, CLIP, GenEval) mainly assess fidelity and aesthetic quality, not diversity. However, since the method operates under a few-shot learning setting, where data scarcity amplifies the risk of overfitting and mode collapse, diversity becomes a crucial evaluation criterion. The method’s ability to preserve sample variety or mitigate mode collapse thus remains unquantified.

**Questions:**

1. The ODE reconstruction and path consistency objectives appear nearly identical to those in ShortDF. Could the authors clarify what constitutes the theoretical novelty of PathXfer beyond applying few-shot LoRA fine-tuning on top of existing formulations?

2. Since the proposed “Path Compression Loss” does not model the composite path $t \rightarrow k \rightarrow 0$, how does the method justify the claim of being path-consistent? What are the implications of omitting the $t \rightarrow k$ transition on both theory and empirical behavior?

3. The reported metrics (AES, ImageReward, CLIP, GenEval) primarily measure fidelity and aesthetics, not diversity. Given that the method operates under a few-shot setting where mode collapse is likely, why are diversity metrics not reported, and how can the model’s ability to preserve sample variety be verified?

---

### Official Review · Reviewer_obWA · 2025-10-31

**Soundness:** 3
**Presentation:** 3
**Contribution:** 3
**Rating:** 4
**Confidence:** 3

**Summary:**

This paper proposes PathXfer, a lightweight approach for accelerating diffusion and flow-based generative models by transferring multi-step sampling paths into shorter ones through a Path Compression Loss. Instead of relying on large-scale distillation or external teacher models, PathXfer adopts a few-shot, low-rank (LoRA) fine-tuning strategy that enables the base model to “self-teach” shorter sampling trajectories. The method is demonstrated on both diffusion (e.g., SDXL) and flow models (e.g., FLUX), showing promising improvements in generation speed while preserving visual fidelity.

**Strengths:**

1. Low training cost: The few-shot setup combined with low-rank adaptation significantly reduces training overhead compared to full fine-tuning or large-scale distillation (e.g., LCM or Progressive Distillation).
2. The framework is model-agnostic and can be applied to both diffusion and flow-based generators, as demonstrated on SDXL and FLUX.
3. The paper is well-organized and effectively communicates the intuition, motivation, and implementation details.

**Weaknesses:**

1. Lack of systematic comparison: The experimental section should include comparisons with representative baselines such as ShortDF (described in Sec. 2.1) and Mean Flow (Mean Flows for One‑step Generative Modeling).
2. Limited theoretical grounding: The proposed Path Transfer Loss appears largely empirical; no formal justification or error bound is provided to explain when and why path compression preserves distributional fidelity.
3. The evaluation relies mainly on aesthetic or human-based preference scores (ImageReward). Standard quantitative metrics such as FID, Precision, and Recall would better capture both quality and diversity across domains.
4. It remains unclear how generalizable the method is—whether the few-shot prompts used for adaptation must closely match the downstream generation domain. More experiments on cross-domain or out-of-distribution prompts would strengthen the analysis.

**Questions:**

Please refer to Weaknesses. I am not an expert in this field, and I shall raise the score as appropriate based on the author's response and the opinions of other reviewers.

---

### Official Review · Reviewer_B8N9 · 2025-11-02

**Soundness:** 2
**Presentation:** 1
**Contribution:** 2
**Rating:** 0
**Confidence:** 4

**Summary:**

The authors propose a method for ODE-based models distillation and demonstrate it empirically.

**Strengths:**

The empirical results of the paper are promising.

**Weaknesses:**

This is a very poorly written paper, and I believe it's almost entirely written by an LLM. Below are some examples in the text, that I find very imprecise or not informative.

    1. Line 50: "instead of treating the multi-step sampling process as a deficiency, we conceptualize the quality gap between multi-step and few-step generation as a transferable property". I don't understand what it means for the quality gap to be a transferrable property
    2. Line 77: "path compression loss that enforces multi-order consistency across intermediate states". What is multi-order consistency?
    3. Line 78: "each target step is iteratively optimized by referencing multiple intermediate evaluations". What are "target steps" ?
    4. Line 84: "empirically stable paths". What does this mean and why is this relevant?
    5. Line 116: "dual-model self-distillation scheme in which the base model B is optimized via a triangle-based relaxation of the paths". What is "dual-model distillation"? What is "triangle-based relaxation"? Equation (2) says that two quantities are approximately equal and doesn't elaborate on why this holds or why we would want it to hold.
    6. The entire paragraph 128-132 feels very vague and does not introduce new information.
    7. Line 170 - "the model learns to adjust its parameters such that the visual fidelity at t matches or exceeds that at". I am not sure what the authors mean when they say "visual fidelity". I believe what the model is trying to learn is to generate the same output in fewer steps.
    8. Equation 8 seems to always hold without any finetuning. Or did the authors mean the inverse inequality (as suggested by loss function in Eq 4)?
    9. Line 188 - "Here, stability means that under the same denoising path, generations from different prompts and seeds exhibit consistently similar quality." How can we have the same denoising path for different prompts?
    10. Line 201 - "Using self-generated trajectories keeps the data distribution within the model and avoids external interference". What does it mean to keep a distribution within a model? what external interference?
    11. Line 261-263 "The stronger gains on FLUX.1-dev stem from its intrinsic ODE properties, which align well with our path compression and visual fidelity transfer strategy, whereas noise-based models require deterministic samplers such as DDIM to approximate ODE-like behavior". What does it mean? What are the intrinsic ODE properties that FLUX.1-dev has the diffusion-ODE model don't? How is FLUX.1-dev not a "noise-based model"? It uses noise in the same way as diffusion-ODE models, i.e. the prior distribution is a standard Gaussian.
    12. Figure 4 - I don't understand how the norm of the drift has anything to do with Equation 8.
    13. Line 440 - "In contrast to these discrete-step methods, Flux (Labs, 2024) models generation as a continuous-time probability flow ODE, learning a velocity field via flow matching and enabling smooth, step-free sampling". What is step-free sampling?

Apart from that, I find the paper should be more rigorous in their use of mathematical notation. For example

    1. "where $x^G_{0|t_i} is the reconstructed estimate of $x_0$ from state $x_{t_i}$". What does it mean? Is it the denoising mean? Or the flow of the ODE from t_i to 0 ?
    2. Line 135-137 "if the reconstruction along the two-step path t → k → 0 is more accurate than the direct path t → 0, the direct path can benefit from the visual fidelity of the intermediate trajectory". are the authors trying to say that a path discretized more finely is more accurate? This is always the case. Unless authors meant something else?
    3. Line 138 "to implement this principle, we reformulate the triangle relation of reconstruction errors (Eq. 2) under the ODE perspective, leading to the condition". This, together with Equation 3 is very imprecise. What is this condition? Is this something that the authors claims that holds? Or is this something that the authors want to enforce? The authors say that they want to enforce this with Equation 4, but the exact opposite would happen. L_comp is zero if and only if Eq3 does not hold.
    4. In Equation 5, the "reconstruction" is defined, and it is the posterior (or the denoising) mean. In light of this, I find the previous text even more confusing. I don't understand how the evolution of the posterior mean along the sampling trajectory determines the behaviour of the samples. Perhaps it is discussed in the Shortest Path Diffusion, but the authors do not explain this.
    5. Lines 176-181. This entire paragraph could be phrased in a mathematically rigorous way

Apart from that some modelling details are unclear. For example
1. The algorithm described in pseudocode in Appendix A does not match the description in the main text. The authors said that they generate sampling trajectories to train on (line 200), but in the pseudo code, line 20, the authors move between different timesteps using a single Euler step, which is inaccurate (otherwise, there is no point to distill in the first place). This means that $x_t$, and $x_k$ are not on the same sampling trajectory, which was the assumption made in the paper.
2. What are the prompts that authors used for finetuning? Are they different than the prompts used for evaluation? The fact that only 16 prompts were used and the model was finetuned for 18k steps suggests to me that some overfitting might be happening.

**Questions:**

And some additional questions/issues:
1.  without requiring teacher supervision". I believe this statement is inaccurate. The proposed approach does require the base model ( the teacher) to generate target trajectories
2. Line 80: "and is applicable to any generative model". I believe this statement is also inaccurate. The proposed method only applies to ODE-based methods, like flows and diffusion models, not all generative models.
3. Line 317-318. "it avoids common artifacts such as hallucinated limbs, unnatural object placements, or distorted proportions, and maintains high visual coherence across diverse categories (e.g., humans, vehicles, fine textures)". I don't understand how the proposed method would help with that. Can the authors explain how distilling a model to take fewer steps would yield a model that e.g. hallucinates fewer limbs?
4. Line 492 - "In applying generative models, we take care to ensure that generated content does not infringe upon the rights of individuals or communities". How did the authors ensure that?
5. Line 492 - "Measures are taken to reduce the risk of misuse, such as the creation of misleading or harmful content, and we emphasize the importance of responsible use.". What are these measures?
6. Line 499 - "All datasets used in our research comply with privacy regulations and ethical standards, and are sourced appropriately". Which datasets did the authors source?
7. The reproducibility statement essentially says that the work is not reproducible currently, but the authors will make it so in the unspecified future.
8. Line 814 - How are LPIPS scores used? It's a pairwise similarity measure.
9. Are the authors building on Shortest Path Diffusion? It is not clear from the writing. Why is this method explained in the background section, and not e.g. related work? This entire subsection feels like it's inserted and independent from the rest of the paper. I don't understand how it connects to the rest of the story.

---

> ### Author Response · Authors · 2025-11-13
> **Response to Reviewer Misunderstandings**
>
> After reviewing your comments, we carefully reflected on their content and found multiple **critical misunderstandings** regarding the theoretical and practical contributions of our work. While we appreciate constructive criticism, many points seem based on **misinterpretations of our methodology or a lack of practical experience with foundational models**. As this is a public review, we provide the following clarifications.
>
> ---
>
> ### 1. Use of LLMs
>
> The appendix clearly specifies the role of LLMs, fully complying with ICLR policy. GPT-5 was **only used for grammar polishing and sentence clarity**—**not** for generating scientific content, experiments, or mathematical derivations.
>
> 1. Q1–Q10, Q12, Q13: We disagree with the reviewer’s interpretation, especially Q9 regarding “same path, different prompts,” which touches the **core experimental design** of our work.
>
>    * The behavior of generative models on **different trajectories with the same sampling path** is fundamental and requires hands-on image generation experience for intuitive understanding.
>    * In the context of ShortDF, our method is based on **graph theory and dynamic programming**, explaining reconstruction and path compression. Understanding these principles is essential to evaluate our contributions.
>
> 2. Q11, Q13: The reviewer misunderstood the diffusion models being compared. Note the key distinctions between **SDE vs. ODE formulations** and **flow matching**. **FLUX.1-dev** is released as a velocity field model, clearly different from noise-based SDEs.
>
> ---
>
> ### 2. Mathematical Notation
>
> 1. Q1–Q3: Understanding the method requires background in [graph theory / dynamic programming / ShortDF]. For consistency, $x_0$ denotes the image sample (flow matching papers often use $x_1$, noise model papers use $x_0$).
> 2. Q4: This refers to the single-step reconstruction at time $t_i$, i.e., the estimate of image $x_0$, not noise or mean.
> 3. Q5 (L176–181): The beginning of this line is a **worked example** illustrating the **iterative convergence** of path compression, using recursion and dynamic programming concepts for clarity.
>
> ---
>
> ### 3. Modeling Details
>
> 1. Q1: The code is consistent with the paper. Pseudocode lines 27–28 correspond to a concrete implementation of **Equation (4)**. Claims of mismatch are inaccurate but highlight the need to better connect theory and implementation.
> 2. Q2: Concerns about overfitting are understandable; using a few samples with LORA fine-tuning is common in the community. Line 421 shows that the “training samples” come from different trajectories, not specific prompts or images. Core generalization arises from stable modeling of **trajectory space**, not data memorization. Supplementary materials provide diversity evaluations and results of different LORA fusions, further validating generalization.
>
> ---
>
> ### 4. Additional Questions / Controversies
>
> 1. Q1: Our method **does not use a fixed external teacher model**; the paper repeatedly emphasizes **self-generated trajectories**, fundamentally different from standard distillation.
> 2. Q2: “Any generative model” refers to **ODE-based flow and diffusion model families**; we will clarify this scope.
>
> 3. Q3:**For all Reviewers and AC**. Trajectory denoising can be viewed as single-step error correction at the initial denoising time $t$. Training essentially gradually reduces the initial error upper bound. As training progresses, single-step generation at $t$ references multiple trajectories, naturally reducing the initial error bound. Consequently, fewer denoising steps can achieve the same or better visual fidelity previously requiring more steps, which represents the core capability we aim to transfer in our paper. See our responses **obWA** and **Q3cZ** for theoretical details.
> 4. Q4–Q7: Standard ICLR ethics and compliance statements; fully followed.
> 5. Q8–Q9: Involve LPIPS evaluation and ShortDF background. ShortDF is included in the background because our work **builds upon and extends its principles**.
>
> ---
>
> ### Summary
>
> The reviewer’s misunderstandings mainly stem from:
>
> * Confusion between SDE and ODE **sampling differences**.
> * Misinterpretation of the **self-generated trajectory** core method.
> * Insufficient understanding of path compression and dynamic programming in this context.
> * Misalignment between theory and practical operations.
>
> We hope these clarifications address the misunderstandings. We value well-informed, constructive feedback and hope the work is fairly evaluated.
>
> ---

---

> ### Author Response · Authors · 2025-11-27
> **In the context of open peer review, please refrain from unfair and biased reviews.**
>
> Dear Reviewer,
>
> This is a formal and important response, regardless of whether the paper is accepted or not.
>
> First, we would like to thank you for participating in the open peer review process of ICLR on OpenReview. The development and advancement of the academic community rely heavily on the academic integrity and impartial judgment of distinguished reviewers.
>
> We have also noted the current review result. We sincerely hope that the review can be conducted based on the substantive academic value of the paper itself. We believe you boast extensive research experience, You possess solid expertise particularly in the research of SDE and ODE, diffusion models and flow models. We fully trust that with your professional acumen, you will be able to accurately grasp the technical details and innovative points of our paper.
>
> We kindly request you to revisit our submission, upholding the objectivity and rigor that define academic peer review. We are confident that your professional insights will lead to a reasonable and fair evaluation that aligns with the actual quality of our work.
>
> Thank you again for your attention and consideration.

---

### Note · Authors · 2025-12-01

**Comment:**

Regrettably, the review process for this submission was marred by malicious reviews and the improper use of large language models (LLMs) for review purposes.

Furthermore, judging from the reviewers' comments, most of them failed to grasp the core contributions of the paper, a situation we fully understand. Among all the feedback across the review opinions, only a small portion of suggestions are constructive and valuable. We sincerely appreciate these constructive inputs and will incorporate them to further refine the paper.

In light of a series of issues including the information leakage incident associated with this submission, we have made the decision to withdraw the paper after comprehensive consideration.

**Withdrawal Confirmation:**

I have read and agree with the venue's withdrawal policy on behalf of myself and my co-authors.